# Gandhi's Militant Nonviolence in the Light of Girard's Mimetic Anthropology

**Wolfgang Palaver** [1,2]

1   Department of Systematic Theology, University of Innsbruck, 6020 Innsbruck, Austria;
    wolfgang.palaver@uibk.ac.at
2   Wallenberg Research Centre at Stellenbosch University, Stellenbosch Institute for Advanced Study (STIAS),
    Stellenbosch 7600, South Africa

**Abstract:** Nuclear rivalry, as well as terrorism and the war against terror, exemplify the dangerous escalation of violence that is threatening our world. Gandhi's militant nonviolence offers a possible alternative that avoids a complacent indifference toward injustice as well as the imitation of violence that leads to its escalation. The French-American cultural anthropologist René Girard discovered mimetic rivalries as one of the main roots of human conflicts, and also highlighted the contagious nature of violence. This article shows that Gandhi shares these basic insights of Girard's anthropology, which increases the plausibility of his plea for nonviolence. Reading Gandhi with Girard also complements Girard's mimetic theory by offering an active practice of nonviolence as a response to violent threats, and by broadening the scope of its religious outreach. Gandhi's reading of the Sermon on Mount not only renounces violence and retaliation like Girard but also underlines the need to actively break with evil. Both Gandhi and Girard also address the religious preconditions of nonviolent action by underlining the need to prefer godly over worldly pursuits, and to overcome the fear of death by God's grace. This congruence shows that Girard's anthropology is valid beyond its usual affinity with Judaism and Christianity.

**Keywords:** nonviolence; M.K. Gandhi; R. Girard; mimetic rivalry; Sermon on the Mount; spiral of violence; detachment; religion; fear of death



## 1. Introduction

The publication of *The Healing Power of Peace and Nonviolence* by the Catholic moral theologian Bernhard Häring in 1986 was a "clarion call to Christians to embrace nonviolent action" (Berger et al. 2020, p. 114). The Catholic Church has reacted positively to this call, as the development of its peace ethics shows. So far, it has culminated in Pope Francis' message for the World Day of Peace on 1 January 2017, in which he mentioned—among other examples of successful practices of nonviolence—Mahatma Gandhi's liberation of India (Francis 2017, #4). Häring's book refers to Gandhi too, recommending his "militant nonviolence" as the "only meaningful and promising way" to fight "oppression and exploitation" (Häring 1986, p. 81). Häring frequently refers also to the work of the French-American anthropologist René Girard, whom he praises for seeing nonviolence as a "central aspect of redemption through Jesus Christ" (Häring 1986, p. 87).

Following Häring's appreciation of Gandhi's militant nonviolence and Girard's mimetic anthropology, this article connects these two approaches, and shows how this leads to a deeper understanding of nonviolence and its religious preconditions. Gandhi shares with Girard important insights into the nature of violence and why nonviolence has become a necessity for the survival of humanity. Reading Gandhi with the eyes of Girard strengthens the plausibility of Gandhi's concept of *satyagraha*. It also complements Girard's mimetic theory by offering an active practice of nonviolence as a response to threats of violence, and by broadening the scope of its religious outreach. Girard's strength is his anthropological analyses of the causes and the nature of human violence. He also examines extensively

how early religions dealt with violence. Like Gandhi, he too, underlines the importance of nonviolence for today. He has, however, less to say about positive and practical ways out of violence in our modern world. Whereas Girard emphasizes a rather passive renunciation of violence, Gandhi offers an active practice of nonviolence that addresses the challenges that Girard's anthropology illuminates. Militant nonviolence represents a third way between complacence that remains indifferent toward injustice and a dangerous imitation of violence that leads to its escalation.

Gandhi's work also complements Girard's anthropology in its religious scope. Shortly after Pope Benedict XVI was elected in 2005, Girard claimed a superiority of Christianity in his support of the Pope's battle against relativism (Girard and Gardels 2005, p. 46). The Indian essayist Pankaj Mishra responded by expressing his admiration for Girard's account of the Gospels as opposing the "cycle of violence", and showing that Gandhi's nonviolence was greatly inspired by the New Testament. With Gandhi he rejected, however, Girard's claims for a Christian superiority, and recommended instead to find—like Gandhi—traditions among the world religions that "chime with the truth of the Gospels" (Mishra 2005, p. 53). With the help of Gandhi, Girard's mimetic anthropology gains in its universal validity because many of his key insights are indeed expressed in other world religions, too.

This article discusses first Girard's and Gandhi's discovery of mimetic rivalry as a cause of violence. The second part deals with the contagious nature of violence and how easily it can escalate. Thirdly, a further step explains how nonviolence can break the cycle of violence, and how Gandhi complements Girard by offering an active understanding of nonviolence. The final part engages with the religious preconditions of nonviolent action by addressing the need to orient our desires in godly pursuits so that we can act in a spirit of detachment, and the need to overcome the fear of death by God's grace.

## 2. Mimetic Rivalry as the Root of Human Conflicts

Girard's basic insight into the roots of human violence is his discovery that many conflicts stem from mimetic rivalries that occur when a person imitates another's desire for an indivisible object. He therefore preferred competition to aggression as the key concept to understand human violence:

> We are *competitive* rather than aggressive. In addition to the appetites we share with animals, we have a more problematic yearning that lacks any instinctual object: *desire*. We literally do not know what to desire and, in order to find out, we watch the people we admire: we imitate their desires. [ . . . ] Unlike animal rivalries, these *imitative* or *mimetic* rivalries can become so intense and contagious that not only do they lead to murder but they also spread, mimetically, to entire communities. (Girard 2004, pp. 9–10)

Girard first discussed mimetic rivalries in personal relationships, often love relationships, as described in key novels in his book *Deceit, Desire, and the Novel* (Girard 1966). In *Violence and the Sacred* he deepened his understanding by turning to sibling rivalries, which are a prevailing theme in ancient and traditional cultures (Girard 1977, pp. 59–67). Following Girard, Jonathan Sacks used recently sibling rivalry as a key to the understanding of religious violence in his careful reading of the Book of Genesis (Sacks 2015). Girard did not limit his reflections to the personal sphere, but also showed how mimetic rivalries contribute to dangerous dynamics in the political realm between states or other political actors, as Girard's last book *Battling to the End* illuminates. By emphasizing competition, Girard was able to reject, for instance, concepts like Huntington's "clash of civilizations" to explain terrorism in our contemporary world. Soon after 9/11, Girard recommended that we focus on globalized competition to explain terrorist attacks against the West, instead of seeing it as being caused by religious differences (Girard and Tincq 2002).[1]

In Gandhi's life and work, we see that he was very much aware of the conflictual potentials that easily follow entanglements of mimetic desire, although he would not put it in these words. Following Girard's unfolding of his understanding of mimetic rivalries,

we can start with Gandhi's marriage to Kasturba. Despite all his efforts to become a perfect husband, he suffered severely from jealousy, which burdened his marriage, as his autobiography tells (Majmudar 2005, pp. 59–60). Girard showed, in his book on Shakespeare, that envy and jealousy are "exactly the same thing", because both are an offspring of mimetic rivalries (Girard 1991, p. 10). Jean-Michel Oughourlian, an Armenian-French psychologist, studied the different forms of jealousy that threaten couples by following Girard's view of mimetic rivalry, and distinguished between jealousy of a third party and jealousy of the other partner (Oughourlian 2010, pp. 119–43). In Gandhi's case, we find a mixture of both forms of jealousy. Reading guides for a good marriage diligently convinced Gandhi of the importance of faithfulness, which he tried to achieve for himself and demanded rigorously from his wife, too:

> The thought made me a jealous husband. Her duty was easily converted into my right to exact faithfulness from her, and if it had to be exacted, I should be watchfully tenacious of the right. I had absolutely no reason to suspect my wife's fidelity, but jealousy does not wait for reasons. I must needs be for ever on the look-out regarding her movements, and therefore she could not go anywhere without my permission. This sowed the seeds of a bitter quarrel between us. The restraint was virtually a sort of imprisonment. (Gandhi 1958–1994, p. 39:14; cf. 25)

Gandhi's struggle with jealousy continued during his years of study in London, and stayed with him for some time in South Africa (Gandhi 1958–1994, p. 39:77). It contributed to his vow of chastity, *brahmacharya*, which he took in 1906 to overcome "lustful attachment" (Gandhi 1958–1994, p. 39:165–71). Oughourlian is right to connect Gandhi's "withdrawing from any carnal possession [ . . . ] to completely liberate love from desire and no longer be dominated by it" with his awareness "that all sexual relations involve a dose of rivalry and aggression that is susceptible of degenerating into violence" (Oughourlian 2010, pp. 68–69). He, however, incorrectly claims that Gandhi and his wife promised to renounce carnal love at their wedding ceremony. Gandhi and Kasturba married at the age of thirteen, and this child marriage was partly responsible for Gandhi's early obsession with sexual love, which even caused him to sleep with his pregnant wife on the night of his father's death. The negligence of his dying father encumbered Gandhi throughout his life (Gandhi 1958–1994, p. 39:28–30). It would, however, be wrong to understand Gandhi's negative view of sexuality in a purely puritanical sense. Erik Erikson shows that at its root was an "aversion against all male sadism—including such sexual sadism as he had probably felt from childhood on to be part of all exploitation of women by men". Gandhi's criticism of his father as being dominated by "carnal pleasures" when he married, at the age of forty-eight, an eighteen-year-old woman—Gandhi's mother—supports Erikson's thesis. According to Uma Majmudar, Gandhi viewed such "old man-young woman marriages [ . . . ] as an abominable social custom that formally sanctioned male violence over females" (Gandhi 1958–1994, p. 39:7; Majmudar 2005, p. 35). It is not by chance that his vow of chastity happened immediately after he served in an ambulance unit during the Bambatha Rebellion, where he witnessed the "outrages perpetrated on black bodies by white he-men" (Erikson 1993, p. 194). Gandhi recognized a close connection between male sexuality and violence (Parekh 1999, pp. 199, 220; Bose 2014, p. 171). After his vow, he described his relation to Kasturba as a relation of friendship in which one no longer regards "the other as the object of lust" (Gandhi 1958–1994, p. 39:222). Gandhi identified sexuality with a narrow but widespread male perspective which, according to Erikson, closes off all possibility that "a sexual relationship could be characterized by what we call 'mutuality'" (Erikson 1993, p. 236). Girard, too, was aware of how easily sexuality can trigger violence: "Sexuality leads to quarrels, jealous rages, mortal combats. It is a permanent source of disorder even within the most harmonious of communities." (Girard 1977, p. 35). Contrary to Gandhi, however, the French-American anthropologist recognizes more clearly mimetic rivalries as the source of violence and knows that, detached from them, sexuality can be enjoyed mutually: "I think that sexual pleasure is possible to the extent that the other is respected—and maybe there's no true satisfaction except in that case, when the shadowy

presence of rivals has been banished from the lovers' bed: that's probably also why it is experienced so rarely." (Girard 2014a, p. 12)

Gandhi was also aware of how strongly mimetic conflicts occur in our daily family lives. He grew up in a closely-linked Indian family in which such conflicts occurred frequently, as his secretary Pyarelal recounted:

> A single tactless remark, a slip or oversight, an uncouth habit, heedlessness or disregard of another's feelings may set people's nerves on edge and make life hell for the whole family. Competition in this narrow world is keen; even the youngsters feel the edge of it; little things assume big proportions; the slightest suggestion of unfairness or partiality gives rise to petty rivalries, jealousies, and intrigues. (Pyarelal 1965, p. 193)

Like Girard, also Gandhi observed sibling rivalry as a root of human conflicts when he remarked on how often fights break out between "two brothers living together" (Gandhi 1958–1994, p. 10:32). He too knew from his own experience that "little quarrels of millions of families in their daily lives" are just part of human life (Gandhi 1958–1994, p. 10:48).

Gandhi's awareness of the mimetic roots of human conflicts becomes most obvious in his book *Satyagraha in South Africa*. In his overview of the history of South Africa, he describes the conflict between the Boers and the English in terms that come close to Girard's mimetic theory in the emphasis on how the proximity between rivals enhances the likelihood of conflicts:

> As the Dutch were in search of good lands for their own expansion, so were the English who also gradually arrived on the scene. The English and the Dutch were of course cousins. Their characters and ambitions were similar. Pots from the same pottery are often likely to clash against each other. So these two nations, while gradually advancing their respective interests and subduing the Negroes, came into collision. (Gandhi 1958–1994, p. 29:16)

Even more interesting from a mimetic point of view is Gandhi's description of the conflicts between Europeans and Indians that broke out in Natal, and his assertion that were also the main cause of Gandhi's plan to stay for one year in South Africa finally becoming twenty-one years. He refers to "competition" to explain the main reason for these conflicts (Gandhi 1958–1994, p. 29:25–28). The other term that he frequently uses to address white discriminations against Indians is "trade jealousy". We find it already in his 1895 petition to Lord Ripon, the colonial secretary in London, and later in a letter to Tolstoy (Gandhi 1958–1994, p. 1:207; 9:444; cf. Coovadia 2020, p. 64). In Transvaal, too, competition made European traders jealous of the Indian newcomers:

> Their great success excited the jealousy of European traders who commenced an anti-Indian campaign in the newspapers, and submitted petitions to the Volksraad or Parliament, praying that Indians should be expelled and their trade stopped. The Europeans in this newly opened-up country had a boundless hunger for riches. (Gandhi 1958–1994, p. 29:31)

It is this longing for wealth that makes the conflict almost inevitable. As Gandhi rightly remarked, the Europeans' aim "to amass the maximum of wealth in the minimum of time" did not allow for Indians to become "co-sharers" with them in South Africa (Gandhi 1958–1994, p. 29:76). Gandhi is aware of the divisiveness of "acquisitive mimesis", i.e., the longing for indivisible goods (Girard 1987, p. 26). Where he reflects on ways to overcome the exploitation of the masses in the Western world, he expresses the need for a just distribution that could not be gained by multiplying "material wants" but by "their restriction consistently with comfort" (Gandhi 1958–1994, p. 28:148): "We shall cease to think of getting what we can, but we shall decline to receive what all cannot get." He also underlines the dangers of acquisitiveness in his interpretation of the *Bhagavadgītā* (4:21; 6:10): "Where there is possessiveness, there is violence", and this necessitates not only the "renunciation of possessions" but also the "desire for possessions too" (Gandhi 1958–1994, p. 32:115, 240; cf. Conrad 2006, p. 217).

Like the current attempts to cloak mimetic conflicts in the terms of a civilizational clash, we find a similar discussion during Gandhi's stay in South Africa. He refers to General Smuts, one of the South African leaders, who described the conflict between Europeans and Indians as a conflict between cultures. According to Gandhi's account of Smuts' position, the General saw neither "trade jealousy or race hatred" as the main problem, but rather deep cultural differences between the "Western civilization" and an "Oriental culture" endangering the Westerners in South Africa, such that they "must go to the wall" as if committing "suicide" (Gandhi 1958–1994, p. 29:77). In a letter to Gandhi, Smuts expressed his position in the following way:

> I may have no racial legislation, but how will you solve the difficulty about the fundamental difference between our cultures? Let alone the question of superiority, there is no doubt but that your civilization is different from ours. Ours must not be overwhelmed by yours. (Hancock 1962, p. 346)

Gandhi saw Smuts as a man of the "highest character among the Europeans", but he nevertheless criticized his position harshly as "hypocrisy" supported by "pseudo-philosophical" arguments seeking a justification to mask selfish enrichment and racism (Gandhi 1958–1994, p. 29:76–77). "The only remaining factors are trade and colour." (Gandhi 1958–1994, p. 29:78). In an article in *Indian Opinion* from February 1905 on "Questions of Colour", Gandhi claims that only a racist view neglects the fact of rivalrous competition: "The origin of the whole matter is trade jealousy. It is this petty motive alone that animates the anti-Indian movement; and it is perfectly apparent to all who are not blinded by colour prejudice." (Gandhi 1958–1994, p. 4:355) By pointing to rivalry as the real cause of the conflicts between Europeans and Indians, Gandhi implicitly deconstructs racism as an offspring of acquisitive mimesis that easily results in scapegoating.[2] He comes close to Girard's insight that racism is best explained with the help of our modern use of the term scapegoat, which describes how groups often contain their internal rivalries by channelling them to the outside. Scapegoats multiply "wherever human groups seek to lock themselves into a given identity—communal, local, national, ideological, racial, religious, and so on" (Girard 2001, p. 160; cf. Reineke 1998, pp. 76–81). Girard discovered in ancient myths, medieval texts, and in the modern world many examples of groups and societies that turn to scapegoating if they are facing a crisis. He refers, for example, to medieval and modern anti-Semitism, and also to the fact of how often "ethnic and religious minorities tend to polarize the majorities against themselves [ . . . ]. In India the Moslems are persecuted, in Pakistan the Hindus" (Girard 1986, pp. 17–18; cf. 48).

Furthermore, Gandhi refers to the mimetic image of the "dog-in-the-manger" to describe the conflicts between Europeans and Indians in Natal and Transvaal. This ancient metaphor describes a dog enviously preventing the oxen or the horse from accessing the fodder for which the dog himself has no use. It illustrates an advanced stage of mimetic rivalry among humans in which the original object that triggered the conflict has been replaced by aggression against the rival. According to Girard, mimetic conflicts start with the rivalry over a concrete object that is often quickly forgotten as soon as the conflict escalates (Girard 2001, p. 22). Gandhi already used this image in 1895 when he visited the Trappist community in Mariann Hill and observed that the Europeans prevented the Indians from developing agriculture in Natal without using it for themselves:

> All over the Colony, the small farms are owned by Indians, whose keen competition gives offence to the white population. They are following a dog-in-the-manger and suicidal policy in so behaving. They would rather leave the vast agricultural resources in the country undeveloped, than have the Indians to develop them." (Gandhi 1958–1994, p. 1:223)

He criticized the dog-in-the-manger policy frequently during his time in South Africa and also later in his Indian fight against the British salt tax (Gandhi 1958–1994, p. 4:26.117.52.349; 43:168). In 1939, he criticized the Kathiawar States in India for their dog-in-the-manger policy, which prevented them from overcoming the drought in an area where there would

be plenty of water if all aimed for the common good. This critique indirectly shows how often scarcity is not caused by nature, but rather artificially created by mimetic rivalry (Dumouchel 2013, pp. 3–96).

When Gandhi wrote about these conflicts in his book *Satyagraha in South Africa*, he had already stayed in India. In 1925, the problems in South Africa increased again, and there were attempts to repatriate Indians. Gandhi criticized the "Class Areas Bill" at the Kanpur Congress in December 1925, and emphasized again trade jealousy as the main problem. This time he recommended a pamphlet by Frederick B. Fisher, an American Bishop of the Methodist Episcopal Church who had just visited South Africa to investigate the situation of Indians there. Against Smuts' expression of a "conflict of the two civilizations", Gandhi again highlights—along with Bishop Fisher—"jealousy" (Gandhi 1958–1994, p. 29:359–60). Frederick Fisher indeed underlined jealousy in his report, and described the fate of the Indians under attack from "white supremacy" as being similar to Jewish scapegoats in "old Russia" (Fisher 1926, pp. 149–50): "There is a strange jealousy on the part of the whites with reference to the prosperity of the browns." (Fisher 1926, p. 149). In Fisher's later book on Gandhi, he describes this trade jealousy impressively:

> What happens when a brown man can afford a Rolls Royce! Is that not a direct insult to those white people who still have to run Fords? There is something wrong somewhere, argued the whites. We must look into this matter. They did! To such purpose that the legislative adoption of a series of anti-Asiatic Acts aggravated the disparity between the whites and browns. Here was social, legal and commercial discrimination. (Fisher 1932, p. 39)

Mimetic rivalry easily results in violence, as the fate of the Indians in South Africa clearly illustrates.

We can also find evidence for Gandhi's awareness of mimetic dangers during his years in India. A striking example is his positive view of the traditional Vedic division of society into four classes (*varnas*), in which he recognized a bulwark against the dangers of envious comparisons. According to Gandhi, this tradition could help to prevent worldly ambitions for riches from displacing much more important religious pursuits. Gandhi's own formula—"Let us not want to be what everyone else cannot be" (Gandhi 1958–1994, p. 35:520)—illustrates his insight that following the profession prescribed according to one's *varna* could avoid "all unworthy competition" (Gandhi 1958–1994, p. 59:320; cf. Bondurant 1988, pp. 167–72; Conrad 1999, p. 406). Gandhi is, in this regard, close to modern interpreters of the fourfold order described in the *Bhagavadgītā* (4:13) who emphasize a functional order of "complementarity and cooperation, and not competition" without, however, overlooking like them the "underlying inequalities" (Rambachan 2019, p. 155). He did not identify the *varnas* with the caste system, and strongly insisted on the equality of all human beings: "Assumption of superiority by any person over any other is a sin against God and man. Thus caste, in so far as it connotes distinctions in status, is an evil." (Gandhi 1958–1994, p. 46:302) His understanding of "prejudice and racial hierarchy in South Africa" inspired his criticism of the discrimination against the untouchables in India (Coovadia 2020, p. 84): "Just as we are treating our brothers here, our kith and kin are being treated as pariahs and Bhangis in South Africa." (Gandhi 1958–1994, p. 32:510).[3]

Mimetic rivalries easily lead to discrimination, persecution, and violence. The necessary response to this question will be addressed in a later section. We must first reflect on the fact that imitation is not only at the root of human violence but also takes hold of violence itself by igniting a destructive cycle of violence.

### 3. The Contagious Nature of Violence and the Danger of Its Escalation

According to Girard, imitative desire easily causes violent conflicts. This, however, is only the beginning of the cycle of violence. As soon as violence starts to dominate human relations, it becomes more and more contagious. Frederick Hacker, a psychiatrist and expert on violence, formulated a thesis about the nature of violence that parallels Girard's insight: "Violence is as contagious as the plague." (Hacker 2017, p. 13; cf. Tournier 1978,

pp. 13–14, 91). Girard already described the contagious nature of violence in his earlier work:

> The *mimetic* attributes of violence are extraordinary—sometimes direct and positive, at other times indirect and negative. The more men strive to curb their violent impulses, the more these impulses seem to prosper. The very weapons used to combat violence are turned against their users. Violence is like a raging fire that feeds on the very objects intended to smother its flames. (Girard 1977, p. 31)

This contagious nature of violence easily results in an escalation to extremes: "The slightest outbreak of violence can bring about a catastrophic escalation." (Girard 1977, p. 20) Girard refers, for instance, to the "nuclear rivalry" that installs the atomic bomb as the world's supreme idol, ultimately leading towards death (Girard 1987, pp. 255, 414).

Girard describes the escalating dynamic of violence even more broadly in his book *Battling to the End*, which reflects on Carl von Clausewitz's theory of war to explain our world of global terrorism and wars against terror. The passage that immediately caught Girard's eye in the work of Clausewitz was the description of war as a mimetic relationship between two rivals that ultimately surges toward extremes: "War is an act of force, and there is no logical limit to the application of that force. Each side, therefore, compels its opponent to follow suit; a reciprocal action is started which must lead, in theory, to extremes." (Clausewitz 1984, p. 77) Girard emphasizes this anthropological insight of Clausewitz over his much more often quoted remark that war is nothing but an instrument of politics. "If you start reading Clausewitz carefully, you can see it works exactly like a mimetic novel. It doesn't matter which side wins. Clausewitz does not teach you how to win, but he constantly shows you the mimetic nature of war." (Haven 2020, p. 107). A good example of the way in which a careful reading of Clausewitz reveals his deeper anthropological insights can be found in the chapter in which he explains war as a means of politics, where he does not overlook the fact that means easily influence and often also change ends. The "political aim" is, according to Clausewitz, not "a tyrant. It must adapt itself to its chosen means, a process which can radically change it" (Clausewitz 1984, p. 87). This confirms Girard's mimetic reading, but also precedes Gandhi's reflection on the necessity that means must correspond with ends (Conrad 2006, p. 85). Girard's careful reading of Clausewitz shows that the underlying mimetic dynamic often does not allow politics to prevent a violent "escalation to extremes" (Girard 2010, pp. 53–57). It is this dangerous escalation that threatens, according to Girard, our modern world.

Gandhi shares Girard's insight into the contagious nature of violence. In his book about Indian self-rule from 1909, *Hind Swaraj*, he distanced himself from terrorist attacks against the British occupation, referencing Jesus' saying about "those that wield the sword shall perish by the sword" (Gandhi 1958–1994, p. 10:48 [Mt 26:52], cf. Chandhoke 2014, pp. 72–81). Gandhi feared that by violently fighting against the occupiers, India would just mirror the occupying power. The Boer War also provided him with an example of escalating violence when he observed that "each side was protesting against the other's activities and strengthening its own preparations" (Gandhi 1958–1994, p. 29:60). He even more clearly expressed the danger of escalating violence in 1947, close to the end of his life, when Hindus and Muslims fought against each other in India. A first quote resonates with Girard's reading of Clausewitz: "Once the evil spirit of violence is unleashed, by its inherent nature it cannot be checked or even kept within any prescribed limits." (Gandhi 1958–1994, p. 87:424). Another quote from 1947 problematizes self-defense that often cannot be distinguished from attack: "Self-defence is invoked for taking up the sword. But I have never known a man who has not passed from defence to attack. It is inherent in the idea of defence." (Gandhi 1958–1994, p. 88:145).

Before Gandhi wrote *Hind Swaraj* on a ship going from England to South Africa, he stayed, in 1909, for some months in London, where he discovered in the *Gujarati* of Bombay a poem that helped him to illustrate the self-destructive and contagious nature of violence.

The first lines of the poem 'Blow for a Blow' show how violence begets violence, and moreover aims at self-destruction like the moth is attracted by the flame:

> The lamp not burning,
> On what will the moth throw itself and be burnt?
> Seeking to burn us,
> You burn yourself first. (Gandhi 1958–1994, p. 9:489)

The moth-and-flame metaphor illustrates the self-destructive nature of violence and highlights the mimetic dimension of it, because it is the fire that attracts the moth like violence begets violence. Gandhi understood very well that all violence, like fire, will ultimately destroy itself, and can only sustain itself if it is fed by counterviolence—in this manner, mutual violence escalates catastrophically. At the time that Gandhi wrote *Hind Swaraj* in 1909, he also translated Tolstoy's *Letter to a Hindoo* into Gujarati. In his introduction to this edition, he emphasizes the importance of it to the understanding of the way in which injustice can be resisted non-violently, and accuses those you do not follow the way of nonviolence of being "caught up in the toils of this huge sham of modern civilization, like moths flitting round a flame" (Gandhi 1958–1994, p. 10:2). He sees modern civilization as it was exemplified by the British Empire, as being governed by the law of brute force. In *Hind Swaraj* he does not directly mention the moth-and flame metaphor, but by mentioning all of the "victims" that are "destroyed in the fire of civilisation" with its "scorching flame", he indirectly refers to it (Gandhi 1958–1994, p. 10:24). Like fire, a society ruled by force is self-destructive, and Gandhi tried to convince Indians to resist the contagious attraction of brute force if they do not want to be destroyed like moths in the flame.

The moth-and-flame metaphor has a long tradition in India. It is mentioned in the *Bhagavadgītā* (11:29), in the *Arthasastra* (7.15.14), and in traditional Indian sayings. One of these sayings compares the rash fight against an enemy with the moth aiming for its self-destruction: "He who through folly, sets out impetuous to face a foe without judging rightly/the other's power and his own, will perish like the moth that flies headlong into the fire." (Sarma 2006, p. 136) Violent revolutionaries sacrificing their lives in their fight against the colonial power behaved like moths aiming for the flame. Gandhi did not directly mention this metaphor when he referred to Madan Lal Dhingra, the student who assassinated a British official in London shortly before Gandhi arrived in 1909, but the words he used in *Hind Swaraj* chime with the metaphor: "Dhingra was a patriot, but his love was blind. He gave his body in a wrong way; its ultimate result can only be mischievous." (Gandhi 1958–1994, p. 10:42). According to Gandhi, this blind love was the result of an intoxication by bad ideas (Gandhi 1958–1994, p. 9:302).[4] Others who followed the example of Dhingra used the moth-and-flame metaphor to recruit more revolutionaries. The most prominent of them was Bhagat Singh, who described in his 1928 series on sacrifices for liberty these "patriotic young men" who gave up their lives as "moths hovering around the flame of liberty" (Lal 2019). Singh himself committed several acts of violence in India, for which he was executed in 1931. He and his executed fellows were widely celebrated as martyrs who, like moths, sacrificed their lives for India's independence (Maclean 2014).

After the discovery of this poem, Gandhi frequently used the moth-and-flame metaphor to describe the self-destructive nature of violence (Gandhi 1958–1994, p. 22:62; 30:372; 53:09–10). The most striking example occurs in Gandhi's letter to Jawaharlal Nehru, which he wrote on 5 October 1945, just two weeks after they discussed the dropping of two atomic bombs on Hiroshima and Nagasaki. According to Gandhi, the atomic bomb is the summit of accelerated technological progress as it is expressed in humanity's reliance on machines and life in cities, against which he recommends the spinning wheel and village life as preconditions for a nation committed to truth and nonviolence:

> It does not frighten me at all that the world seems to be going in the opposite direction. For the matter of that, when the moth approaches its doom it whirls round faster and faster till it is burnt up. It is possible that India will not be able to

escape this moth-like circling. (Gandhi 1958–1994, p. 81:319–20; cf. Rothermund 1998, pp. 107–17)

Gandhi saw the atomic bomb as the peak of the Western reliance on brute force, and did not overlook its mimetic dimension when he claimed, in an interview on the day before he was assassinated, that the United States should give up nuclear weapons because "the war ended disastrously and the victors are vanquished by jealousy and lust for power" (Gandhi 1958–1994, p. 90:522). He not only shares with Girard the recognition of the dangers coming along with nuclear rivalry but also recognizes that the bomb "usurps the place of God" (Gandhi 1958–1994, p. 88:167).

### 4. Breaking with the Cycle of Violence: Progressively Substituting Force with Nonviolence

Girard's recognition of an apocalyptic escalation of violence made him understand the urgency to renounce violence and retaliation, as recommended by the Sermon on the Mount: "The definitive renunciation of violence [ . . . ] will become for us the condition *sine qua non* for the survival of humanity itself and for each one of us." (Girard 1987, p. 137; cf. pp. 97, 258; 1991, p. 282; 2014b, p. 20). Reflecting on Jesus' demand to overcome retaliation and renounce violence in Mt 5:38–40 and in Lk 6:33–35, Girard shows its plausibility from a mimetic perspective, even questioning self-defence, much like Judith Butler in her recent book on nonviolence:

> To leave violence behind, it is necessary to give up the idea of retribution; [ . . . ] we think it quite fair to respond to good dealings with good dealings, and to evil dealings with evil, but this is precisely what all the communities on the planet have always done, with familiar results. People imagine that to escape from violence it is sufficient to give up any kind of violent *initiative*, but since no one in fact thinks of himself as taking this initiative—since all violence has a mimetic character, and derives or can be thought to derive from a first violence that is always perceived as originating with the opponent—this act of renunciation is no more than a sham, and cannot bring about any kind of change at all. Violence is always perceived as being a legitimate reprisal or even self-defence. So what must be given up is the right to reprisals and even the right to what passes, in a number of cases, for legitimate defence. Since the violence is mimetic, and no one ever feels responsible for triggering it initially, only by an unconditional renunciation can we arrive at the desired result. (Girard 1987, p. 198; cf. Butler 2020, pp. 51–55)

Girard justly highlights the renunciation of counterviolence in the Sermon on the Mount. A masochistic quietism is not recommended by Jesus, but rather a retreat from imitating violence: "When Christ says 'if someone strikes you on the right cheek, turn to him the other also' (Matt. 5:39) he is not advocating a form of masochistic quietism, but the danger of bad reciprocity, of any escalation of bad mimesis" (Girard 2008, pp. 252–53). By reading Mt 5:39 in connection with Ps 37:8, the Jewish theologian Pinchas Lapide translates this verse in a way that highlights its rejection of mimetic counterviolence: "Do not compete in doing injustice." (Lapide 1986, p. 134). Although Girard rejected criticisms of the Sermon on the Mount "as a utopian sort of pacifism, manifestly naïve and even blameworthy because servile, doloristic, perhaps even masochistic", he did not become a pacifist himself (Girard 2014b, p. 19): "I should make it clear that I myself am not an unconditional pacifist, since I do not consider all forms of defense against violence to be illegitimate." (Girard 2014b, p. 131). In *Battling to the End*, he even claims—along with Carl Schmitt—that "pacifism fans the fires of warmongering" (Girard 2010, p. 65). For a concrete example, he refers to the fact that France did not react against Hitler's re-arming of the Rhineland in 1936 when it could have stopped Hitler's career immediately, and most likely for ever (Girard 2010, pp. 182–88). Girard, however, did not develop a peace ethics or a political ethics. He remained quite vague in this regard, and often referred to the religious conclusions that he drew from his anthropological insights. When he was asked in 2005 what he would

propose to politicians following his understanding of Clausewitz, he evaded the question: "It's a complicated question because my vision fundamentally is religious. I believe in non-violence, and I believe that the knowledge of violence can teach you to reject violence." (Haven 2020, p. 107) Despite Girard's rejection of quietist readings of the Sermon on the Mount, he was not able to move beyond a rather passive renunciation of violence. This becomes most obvious in his last book *Battling to the End*, which recommends Hölderlin's "mystical quietism" (Girard 2010, p. 123). Several authors who are familiar with mimetic theory have criticized Girard for his quietist leanings (Reineke 2012; Colborne 2013; Avery 2013, pp. 244–50). Furthermore, his interpretation of Mt 5:39 also remains quite passive, as the following passage shows, where Girard explains the rules of the Kingdom of God as the request to end the mimetic rivalry by giving "way completely to your rival" (Girard 2014a, p. 47; Cayley and Girard 2019, pp. 48, 50): "If you've been hit on the left cheek, offer up the right." This rather passive interpretation does not really grasp the gist of the Sermon on the Mount, as we immediately can recognize in Girard's mixing up of the right cheek that is mentioned by Jesus (Mt 5:39: "if anyone strikes you on the right cheek, turn the other also") with the left one. At first sight, this does not seem to matter much, but we will soon see that it indicates his neglect of the active side of nonviolent resistance.

Girard's appreciation of the Sermon on the Mount brings him close to Gandhi, who he once remarked was influenced by Jainism and Christianity, and that "he opted for the kind of political action that is more compatible with the latter" (Girard 2008, p. 212). Gandhi, indeed, shares with Girard this focus on the New Testament, but goes further than the French American anthropologist in his ethical and political attempts to practice an active nonviolence. The Girardian psychologist Oughourlian rightly mentions Gandhi as a historical example of a politician who understood the mimetic dynamics and opted, for this reason, for nonviolent action close to the Sermon on the Mount (Oughourlian 2012, pp. 66–67).

Like Girard, Gandhi was a great admirer of the Sermon on the Mount. Christians in London introduced him to the New Testament, and to the work of Leo Tolstoy, which gave him "faith in non-violence" (Gandhi 1958–1994, p. 37:261–62). In his autobiography, he expresses his admiration of the Sermon on the Mount by underlining those verses that also caught Girard's attention, and by showing how it aligns with insights in other religious traditions:

> The verses, 'But I say unto you, that ye resist not evil: but whosoever shall smite thee on thy right cheek, turn to him the other also. And if any man take away thy coat let him have thy cloak too', delighted me beyond measure and put me in mind of Shamal Bhatt's 'For a bowl of water, give a goodly meal', etc. My young mind tried to unify the teaching of the *Gita*, *The Light of Asia* and the Sermon on the Mount. (Gandhi 1958–1994, p. 39:61)[5]

These verses from the Sermon on the Mount were important for Gandhi because, very similarly to Girard, he knew about the mimetic dynamic of violence and its dangerous escalation. In 1924, he noted that traditional wisdom was aware that the mirroring of violence must be stopped in order to overcome it:

> It has been my invariable experience that good evokes good, evil—evil; and that therefore, if the evil does not receive the corresponding response, it ceases to act, dies of want of nutrition. Evil can only live upon itself. Sages of old, knowing this law, instead of returning evil for evil, deliberately returned good for evil and killed it. Evil lives nevertheless, because many have not taken advantage of the discovery, though the law underlying it acts with scientific precision. (Gandhi 1958–1994, p. 24:55)

Like Girard, who claimed for his mimetic anthropology a scientific objectivity, Gandhi also talks about a natural law that must be understood in order to overcome violence. The most important ethical conclusion that Gandhi draws from his insight into mimetic dynamics was his insistence that the means to achieve peace must be nonviolent. Against

the wide-spread belief that ends justify means, Gandhi emphasizes that the means must correspond with the end. In *Hind Swaraj* he rejects "brute force" as the adequate means to end the British occupation of India, because one cannot achieve a lasting peace by sowing war: "The means may be likened to a seed, the end to a tree; and there is just the same inviolable connection between the means and the end as there is between the seed and the tree." (Gandhi 1958–1994, p. 10:43). He often repeated this insight in his writings. Another example can be found in 1924, when he rejected Bolshevism for its belief in "short-violent-cuts to success" (Gandhi 1958–1994, p. 25:424): "Those Bolshevik friends who are bestowing their attention on me should realize that however much I may sympathize with and admire worthy motives, I am an uncompromising opponent of violent methods even to serve the noblest of causes."

Tolstoy's writings not only strengthened Gandhi's faith in nonviolence but also led him to an active interpretation of the Sermon on the Mount. Tolstoy's emphasis on non-resistance did not mean to remain indifferent about evil: "Non-resistance to evil has nothing to do with tacit acceptance of the phenomena of evil. Tolstoy was never averse to fighting evil by every moral instrument at his command, and hurling his massive protests into the world." (Nigg 1962, p. 391). John Howard Yoder rightly noticed that "nonretaliation" should be preferred to "nonresistance" to describe Tolstoy's attempt to break the chain of evil (Yoder 2009, p. 55). Tolstoy talked frequently about the "doctrine of non-resistance to evil by violence" in his book *The Kingdom of God is Within You*, enhancing the usual translations of Mt 5:39 by adding the term "violence" (Tolstoy 2010, p. 5). Gandhi followed Tolstoy when he criticized European Christians for identifying Jesus with "passive resistance" (Gandhi 1958–1994, p. 90:129): "As I read the New Testament for the first time I detected no passivity, no weakness about Jesus as depicted in the four gospels and the meaning became clearer to me when I read Tolstoy's *Harmony of the Gospels* and his other kindred writings." Gandhi's reading of the New Testament contributed to his concept of nonviolence, which was neither indifferent to evil nor avoided conflicts. In his book *Satyagraha in South Africa*, he underlines the active nonviolence that he recognized in Jesus Christ: "Jesus Christ indeed has been acclaimed as the prince of passive resisters but I submit in that case passive resistance must mean satyagraha and satyagraha alone." (Gandhi 1958–1994, p. 29:96). Gandhi's view of Jesus has become an inspiration for Christians to gain an active interpretation of the Sermon on the Mount.

A careful reading of the verse that recommends the turning of the other cheek will recognize that Jesus talks about being slapped on the right cheek. This, however, is only possible if the offender uses the back of his right hand. Influenced by Gandhi, the biblical scholar Walter Wink explains how a back-handed slap is a humiliating blow by which masters insulted slaves, men insulted women, and Romans insulted Jews (Wink 1999, pp. 101–3; cf. Lapide 1986, pp. 121–27). To turn the other cheek means to insist on being recognized as an equal in the confrontation. It refuses to be humiliated without, however, retaliating. Jesus himself questioned the police who struck him on the face during his interrogation by the high priest: "If I have spoken wrongly, testify to the wrong. But if I have spoken rightly, why do you strike me?" (Jn 18:23). Wink links Jesus' words about turning the other cheek with Gandhi's "first principle of nonviolent action" demanding "non-co-operation with everything humiliating" (Wink 1999, p. 102; Gandhi 1958–1994, p. 83:206). The Sermon on the Mount does not recommend a passive acquiescence but asks for an active engagement with evil without imitating it. In seminars in South Africa in 1986—during apartheid—Wink showed that Jesus offers a "third way" beyond "flight or fight" by opposing evil without mirroring it (Wink 1987, p. 23). Like Gandhi, Wink also claims that "Jesus abhors both passivity and violence as responses to evil" (Wink 1987, p. 14). Wink calls Jesus' third way "militant nonviolence", and distinguishes it from "passivity" as well as "violent opposition" (Wink 1987, p. 13; cf. Wink 1999, p. 143). He therefore translates Mt 5:39 like Tolstoy, by underlining the renunciation of violence: "Do not violently resist evil." (Wink 1987, p. 22; cf. Wink 1992, pp. 184–89; Wink 1999, pp. 101, 22; Wink 2003, pp. 11, 27).

A key text for Gandhi's active understanding of nonviolence is Tolstoy's *Letter to a Hindoo*. In the introduction to his Gujarati translation of it, he shows that Tolstoy's rejection of retaliation "does not mean [ . . . ] that those who suffer must seek no redress" (Gandhi 1958–1994, p. 10:1). According to Tolstoy, it is also essential not to submit to injustice. This becomes even more explicit in Gandhi's English introduction to this letter, in which he quotes the following sentence from it: "Do not resist evil, but also yourselves participate not in evil, in the violent deeds of the administration of the law courts, the collection of taxes and, what is more important, of the soldiers, and no one in the world will enslave you" (Gandhi 1958–1994, p. 10:4; cf. Bartolf 1997, p. 26). Tolstoy's warning against taking part in evil influenced Gandhi's nonviolent struggle against the colonial rule of the British Empire, as his non-cooperation movement showed, which he launched in 1920 (Guha 2018, pp. 133–60). When Christians challenged him on the supposition that his non-cooperation campaign goes against Christ's saying that one has to "render unto Caesar, the things which are Caesar's" (Mt 22:21), Gandhi responded that Christ expressed with these words the "great law [ . . . ] of refusing to co-operate with evil" (Gandhi 1958–1994, p. 23:105; cf. 107, 43:31).

Vanessa Avery criticized Girard for overlooking the fact that the Sermon on the Mount demands an active nonviolence, and not primarily a self-sacrificial attitude. Following Wink, she refers to Rosa Parks, the black seamstress whose refusal to give up her seat in the bus led to the Montgomery bus boycott, the foundational event in the civil rights movement in the United States (Avery 2013, p. 249; King 2010, pp. 30–34). This movement was inspired by Gandhi and his understanding of Jesus, as Martin Luther King Jr. observed: "Gandhi was probably the first person in history to lift the love ethic of Jesus above mere interaction between individuals to a powerful and effective social force on a large scale" (King 2010, p. 84). King coined the term "militant nonviolence", which matches—as Erik Erikson very well understood—with Gandhi's understanding of *satyagraha* (King 1991, pp. 348, 483–84; Erikson 1993, p. 197; Colaiaco 1988). Judith Butler follows this militant understanding of nonviolence that "must be aggressively pursued", and which Albert Einstein called "militant pacifism" (Butler 2020, pp. 20, 27). Gandhi's active view also explains why he endorsed not only the negative withholding of violence but also the active pursuit of social justice: "No man could be actively non-violent and not rise against social injustice no matter where it occurred." (Gandhi 1958–1994, p. 71:424; cf. Rambachan 2015, p. 105).

Gandhi did not advocate an absolute pacifism (Chandhoke 2014, pp. 92–94; Parel 2016, pp. 106–11; Jahanbegloo 2021, pp. 65–80). Closer to Girard than to Tolstoy, he was aware that the fetishization of nonviolence could result in counterproductive consequences (Steffen 2007, pp. 134–80). He remarked already in *Hind Swaraj* that preventing a child "forcibly [ . . . ] from rushing towards the fire" cannot be understood as an act of violence (Gandhi 1958–1994, p. 10:46). Some years later, he discussed the example of a man running amok, and said that he might be killed to protect the community. According to Gandhi, it is the intention that decides if such an act is violent or nonviolent: "The fact is that ahimsa does not simply mean non-killing. *Himsa* means causing pain to or killing any life out of anger or from a selfish purpose, or with the intention of injuring it. Refraining from so doing is ahimsa." (Gandhi 1958–1994, p. 31:544; cf. 24:379–80; 37:298, 310–13). In 1918, he defended in a letter to his friend C.F. Andrews his recruiting of Indian soldiers for the British Empire:

> Under exceptional circumstances, war may have to be resorted to as a necessary evil [ . . . ]. If the motive is right, it may be turned to the profit of mankind and that an ahimsaist may not stand aside and look on with indifference but must make his choice and actively co-operate or actively resist. (Gandhi 1958–1994, p. 14:477; cf. 37:269–71)[6]

Gandhi would concur with Girard's criticism of France's pacifistic reluctance to stop Hitler in 1936 because it was not nonviolence out of strength, as he understood his concept of *satyagraha*, but a cowardly attitude that is worse than violence (Gandhi 1958–1994, p. 18:132).

Similarly, he criticized the Munich agreement with Hitler as a "peace without honour" (Gandhi 1958–1994, p. 67:404). Gandhi completely rejected the persecution of the Jews by the Nazis, and claimed that "if there ever could be a justifiable war in the name of and for humanity, a war against Germany, to prevent the wanton persecution of a whole race, would be completely justified" (Gandhi 1958–1994, p. 68:138). He himself, however, did not believe in war, and recommended in 1938 that the Czechs and the Jews should fight non-violently against Hitler (Gandhi 1958–1994, p. 67:404–6; 68:137–39; cf. Guha 2018, pp. 54–60, 550–52).[7] This recommendation has been discussed since, and is indeed questionable (Meir 2021), especially with hindsight. After Hitler unleashed the war, Gandhi moved towards a more qualified understanding of nonviolence that also allowed violent resistance in specific circumstances to count as a form of resistance that is close to the ideal of nonviolence. After Hitler's troops invaded Poland in 1939, Gandhi recognized that, in this situation, only a violent self-defence was available for this country: "If Poland has that measure of uttermost bravery and an equal measure of selflessness, history will forget that she defended herself with violence. Her violence will be counted almost as non-violence." (Gandhi 1958–1994, p. 70:181). He defended his position regarding an "almost non-violence" in later discussions. In August 1940, he summarized his revised view in the following way:

> If a man fights with his sword single-handed against a horde of dacoits armed to the teeth, I should say he is fighting almost non-violently. Haven't I said to our women that, if in defence of their honour they used their nails and teeth and even a dagger, I should regard their conduct non-violent? She does know the distinction between *himsa* and ahimsa. She acts spontaneously. Supposing a mouse in fighting a cat tried to resist the cat with his sharp teeth, would you call that mouse violent? In the same way, for the Poles to stand valiantly against the German hordes vastly superior in numbers, military equipment and strength, was almost non-violence. (Gandhi 1958–1994, p. 72:387–88; cf. 433–34, 74:368, 75:38, 77:146)

This more balanced view of nonviolence allows Adam Roberts to associate Gandhi—like Martin Luther King—with a concept that he calls "progressive substitution". Force has, according to this concept, an important function in policing and defence as long as it cannot be substituted by nonviolent means: "In this view, civil resistance needs to be developed skilfully and strategically if it is to serve the functions previously served by armed force. The hope is that it will replace reliance on force progressively in a succession of issue-areas. The central idea is that only if there is a viable substitute can force be effectively renounced." (Roberts and Ash 2009, p. 8).

To provide evidence for Roberts' thesis we can refer, for instance, to Gandhi's support of the Khilafat movement in 1920, when he joined Indian Muslims in their political campaign to restore the Ottoman Caliphate. When he was asked if this alliance contradicted his commitment to nonviolence, he admitted that this movement would indeed defend Islam "by the sword" (Gandhi 1958–1994, p. 20:165). This, however, did not foreclose his support:

> A believer in non-violence is pledged not to resort to violence or physical force either directly or indirectly in defence of anything, but he is not precluded from helping men or institutions that are themselves not based on non-violence. If the reverse were the case, I would, for instance, be precluded from helping India to attain swaraj because the future Parliament of India under swaraj [ . . . ] will be having some military and police forces.

He was aware that an independent India would rely on nonviolence only to a certain degree. Gandhi's reflections on how a state should be organized prove Roberts' thesis, too. Contrary to Tolstoy's anarchism, Gandhi did not reject the state completely, but saw a certain need of it. In December 1921, he responded to the question of whether imprisoned *satyagrahis* should refuse to do any work in the prisons. He rejected that position because

he did not foresee a society without prisons and warned of "chaos and anarchy", claiming that a "civil resister is [ . . . ] a friend of the State" (Gandhi 1958–1994, p. 22:19). This positive view of the state, however, should not overlook the fact that Gandhi was, in general, closer to anarchism than to a full endorsement of the modern concept of the state with its coercive means (Gandhi 1958–1994, p. 13:214). According to Gandhi, a "non-violent State will be an ordered anarchy", asking—like Henry David Thoreau—for a state "which is governed the least" (Gandhi 1958–1994, p. 72:388–89; cf. 47:91; Marin and Blume 2019; Thoreau 2013, p. 145). This minimalist view of the state results in Gandhi's idea about the type of police force which is appropriate for a state that is committed to nonviolence. In a discussion with pacifists in February 1940, he remarked that a government "cannot succeed in becoming entirely non-violent, because it represents all the people. I do not today conceive of such a golden age". For this reason, he maintained that "even under a Government based primarily on non-violence a small police force will be necessary" (Gandhi 1958–1994, p. 71:226; cf. 72:388–89; Parel 2016, p. 110). A couple of months later he published his "idea of a police force" that is highly relevant for our world of today if we think of all the cases of police violence:

> The police of my conception will [ . . . ] be of a wholly different pattern from the present-day force. Its ranks will be composed of believers in non-violence. They will be servants, not masters, of the people. The people will instinctively render them every help, and through mutual co-operation they will easily deal with the ever-decreasing disturbances. The police force will have some kind of arms, but they will be rarely used, if at all. In fact the policemen will be reformers. (Gandhi 1958–1994, p. 72:403)

On the international level, Gandhi hoped to substitute armies with an "international police force", as he recommended it in a statement on the occasion of the San Francisco Conference preparing the Charta of the United Nations in 1945 (Gandhi 1958–1994, p. 79:389–91).

## 5. Overcoming Mimetic Rivalry by Opening Up to God and Losing the Fear of Death

Judith Butler asks in her book on nonviolence for "an egalitarian imaginary that apprehends the interdependency of lives" (Butler 2020, p. 184; cf. Du Toit and Vosloo 2021). Gandhi represents such an alternative imaginary with his call for a universal fraternity that includes all living creatures. A religiously motivated "embrace all life" is at the centre of this universalism: "I want to realize brotherhood or identity not merely with the beings called human, but I want to realize identity with all life" (Gandhi 1958–1994, p. 40:109). Tolstoy influenced Gandhi's understanding of universal fraternity by recognizing as the first principle of the "Gospel teaching [ . . . ] that all alike are sons of God and therefore brothers and equals" (Tolstoy 2010, p. 129). From Tolstoy's view of fraternity follows his rejection of violence (Conrad 1999, pp. 398–99). Even more important was Gandhi's belief in the Hindu doctrine of *advaita*, non-dualism, that "teaches that the human self [ . . . ] is not different from the limitless *brahman* and is present identically in every being" (Rambachan 2015, p. 10; Chandhoke 2014, pp. 90–91). Gandhi expressed his belief in *advaita* in the following way: "I believe in the essential unity of man and for that matter of all that lives. Therefore I believe that if one man gains spiritually, the whole world gains with him and, if one man falls, the whole world falls to that extent." (Gandhi 1958–1994, p. 25:390). The ethical principle of nonviolence (*ahimsa*) follows from the *advaita* teaching about the identity and unity of existence (Rambachan 2015, p. 104). A good example of Gandhi's ontological foundation of nonviolence can be found in the poem 'Blow for a Blow', the first lines of which were discussed above. The lines that follow express its *advaita* view:

> The union of soul and body,
> The same in you as in me;
> Unless you wound yourself,
> Us you cannot hurt.
> So soon as I owned myself your lover,
> You stood declared my beloved;

> A name I've bestowed on you,
> And will cease only when I perish.
> Such airs you give yourself today,
> Your eyes stern and proud;
> These your arrows
> Will turn back upon you, myself unharmed.
> You live, if I live; if I die,
> Tell yourself you die too;
> [ . . . ]
> Your being is wrapped up in mine
> Aiming a blow at me,
> You shall only hurt yourself." (Gandhi 1958–1994, p. 9:489–90)

This poem expresses the fundamental unity and identity of all beings, which turns all insults against someone else into self-harm.

Gandhi shared with Tolstoy and Martin Luther King a "cosmically based" rejection of violence, and was deeply convinced that the whole universe is governed by love (Yoder 2009, p. 62). He, however, did not underestimate how often violence occurs in our world. As we saw above, he was aware of how easily brothers can turn into hostile rivals, and he did not overlook that even "cannibalism" occurred among humans (Gandhi 1958–1994, p. 63:321). Despite these harsh realities, he was nevertheless convinced that "love [ . . . ] is the law of life" (Gandhi 1958–1994, p. 63:321). In *Hind Swaraj*, he remarks that history tends to focus on conflicts, and often overlooks reconciliation and peace. Examples of fratricide are well known, and find their way into the news and into history books. However, what about brothers who have overcome their rivalries? "Two brothers quarrel; one of them repents and re-awakens the love that was lying dormant in him; the two again begin to live in peace; nobody takes note of this." (Gandhi 1958–1994, p. 10:48). This trust in the God-given law of love is also the basis for Gandhi's conviction that we must choose nonviolent means: "We are merely the instruments of the Almighty Will, and are therefore often ignorant of what helps us forward and what acts as an impediment. We must thus rest satisfied with a knowledge only of the means and, if these are pure, we can fearlessly leave the end to take care of itself." (Gandhi 1958–1994, p. 29:253).

Gandhi comes very close to Girard's religious reflections on ways out of violence. Both understand that human desire is at the root of violence, and is in need of our careful attention. Girard recognizes in Jesus a model for positive mimesis who does not lead his followers in mimetic rivalries because, like his heavenly father, he does not desire "greedily, egotistically" (Girard 2001, p. 14). Gandhi also refers to examples of positive mimesis that do not end up in deadly confrontations. His spiritual mentor Raychandbhai, who overcame all possessiveness and greed, and whom Gandhi tried to imitate in this respect, is one example. Besides the destructive mirroring of others, Gandhi also knew how positive attitudes can spread mimetically: "If I pull one way, my Moslem brother will pull another. If I put on a superior air, he will return the compliment." (Gandhi 1958–1994, p. 10:30).

In order to understand positive mimesis more deeply, we must explain its religious precondition. We can start with Girard's reading of the tenth commandment in the Hebrew Bible as outlawing mimetic rivalry (Girard 2001, pp. 7–9): "You shall not covet your neighbor's house; you shall not covet your neighbor's wife, or male or female slave, or ox, or donkey, or anything that belongs to your neighbor" (Ex 20:17). In order to resist this acquisitive type of desire, the prohibition to covet the objects of those we imitate is, however, not enough. We need to orient our desire toward objects that do not immediately cause divisions among us and should therefore follow the first three commandments of the Decalogue in the Bible. This means to orient us as much as possible toward God without replacing God idolatrously with temporal goods. We can find this insight in the Hebrew Bible as well as in the New Testament (Deut 6:4–9; Mt 22:37–39). The Christian tradition therefore referred to God as the highest good for our deeper longings. If God is our *summum bonum*, we can imitate each other without automatically becoming enemies, because God

is not a good that is lessened if more people reach out for it. The longing for God can be shared and imitated without being driven into violent relationships. The Christian tradition was fully aware of this important path to overcome violence, as demonstrated if we investigate the writings of Augustine, Thomas Aquinas or Dante and their distinction between temporal and eternal goods. Influenced by Girard, Charles Taylor follows this tradition: "The only way fully to escape the draw towards violence lies somewhere in the turn to transcendence, that is, through the full-hearted love of some good beyond life." (Taylor 2007, p. 639).

Gandhi is an important example to prove that this insight is not only part of the Western or Christian tradition, but forms the core of all world religions. His love of religion was not so much focused on specific religions in their concrete institutional settings, but on the "religion which underlies all religions" (Gandhi 1958–1994, p. 10:24). Where he describes the main teaching of this religion underlying all religions, he comes very close to the Christian teachings about the *summum bonum*:

> Hinduism, Islamism, Zoroastrianism, Christianity and all other religions teach that we should remain passive about worldly pursuits and active about godly pursuits, that we should set a limit to our worldly ambition, and that our religious ambition should be illimitable. Our activity should be directed into the latter channel. (Gandhi 1958–1994, p. 10:24)

Gandhi is aware that nonviolence requires "a living faith in God", and mentions it on top of the qualifications that he recommended that members of the Peace Brigade will need to deal with communal riots (Gandhi 1958–1994, p. 67:126; cf. 66:405–7; Häring 1986, p. 127): "A non-violent man can do nothing save by the power and grace of God."

We can deepen this insight into the relationship between human desire and religious longing by focusing on Gandhi's understanding of his Hindu tradition. Among his most important readings was the *Bhagavadgītā*, of which he used the last verses of the second chapter (*Bhagavadgītā* 2:55–72) for his daily meditations and as key for the whole text (Gandhi 1958–1994, p. 49:485; cf. Chatterjee 1983, p. 113). These verses describe a man of steadfast wisdom (*sthitaprajna*) who is beyond all cravings and liberated from passion, fear, and anger (*Bhagavadgītā* 2:54–55). For Gandhi, his friend and mentor Raychandbhai, a Jain and a trader of pearls and diamonds, was a perfect model in this sense because his life did not revolve around these worldly goods, but he rather longed passionately "to see God face to face" (Gandhi 1958–1994, p. 39:75). The following verses underline Gandhi's understanding of the *Bhagavadgītā*[8]: "In a man brooding on objects of the senses, attachment to them springs up; attachment begets craving and craving begets wrath." (2:62); "The man who sheds all longing and moves without concern, free from the sense of 'I' and 'mine'—he attains peace." (2:71).

Fasting is one of the traditional means to curb desire. The following verses in the *Bhagavadgītā*, however, explain that renunciation as such is not able to curb our cravings. Only our opening up toward the highest good can provide real and lasting peace: "And when, like the tortoise drawing in its limbs from every side, this man draws in his senses from their objects, his understanding is secure." (2:58); "When a man starves his senses, the objects of those senses disappear from him, but not the yearning for them; the yearning too departs when he beholds the Supreme." (2:59: cf. 3:27–28).

In Gandhi's comment on these verses, he claims that self-mortification and fasting are of limited use. True liberation relies on God's grace. We must discover God in our own hearts in order to submit to him faithfully: "One who thus looks upon Me as His goal and surrenders his all to Me, keeping his senses in control, is a yogi stable in spirit." (Gandhi 1958–1994, p. 49:114). It is the task of human beings to become a vessel for God's grace: "And once the grace of God has descended upon him, all his sorrows are at an end. As snow melts in the sunshine, all pain vanishes when the grace of God shines upon him and he is said to be stable in spirit." (Gandhi 1958–1994, p. 49:116).

Close to his interpretation of the *Bhagavadgītā*, and pointing in the same direction, is Gandhi's most important mantra, which he took from the first verse of the *Isha Upanishad*,

and in which he recognized the core of Hinduism: "All this that we see in this great Universe is pervaded by God. Renounce it and enjoy it. Do not covet anybody's wealth or possession." (Gandhi 1958–1994, p. 64:259; cf. 58–60, 89–90).

Gandhi meditated frequently on this mantra. In it, he discovered the religious basis for a "peace with all that lives" and a "universal brotherhood—not only brotherhood of all human beings, but of all living beings" (Gandhi 1958–1994, p. 64:260, 90). From a biblical perspective, this mantra is somewhat analogous to the Decalogue. Its last part parallels the tenth commandment, and its beginning summarizes to some degree the first table of the Decalogue. The full surrender to God liberates us from being pushed to covet mimetically our neighbour's belongings.

Gandhi's emphasis on renunciation does not mean to abstain from active engagement in the world, but to act in a spirit that is free from worldly attachments. He does not withdraw like a "cave-dweller", but toils instead in the "service of my country and therethrough of humanity" (Gandhi 1958–1994, p. 23:349). Acting in the world, however, presupposes detachment: "A follower of the path of renunciation seeks to attain it not by refraining from all activity but by carrying it on in a perfect spirit of detachment and altruism as a pure trust." (Gandhi 1958–1994, p. 37:385–86). Detachment is especially important regarding possessions, but must guide all acting in the world. This attitude has a parallel in Saint Paul's first letter to the Corinthians, in which he calls Christians to relate to property in a spirit of "having as if we have not", and to deal with the world as if one has no dealings with it (1 Cor 7:30–31; Conrad 2006, p. 216).

Gandhi describes his detached acting in the world as "striving for the Kingdom of Heaven which is *moksha*" (Gandhi 1958–1994, p. 23:349). Catholic social teaching describes a quite-similar attitude in its frequent calls to orient oneself toward God as the highest good—the *summum bonum*—and by repeatedly quoting a relevant verse from the Sermon on the Mount: "But strive first for the kingdom of God and his righteousness, and all these things will be given to you as well." (Mt 6:33). Gandhi, too, loved this verse, which he discovered in Tolstoy's *The Kingdom is Within You*, and he referred to it frequently (Gandhi 1958–1994, p. 64:119; Tolstoy 2010, pp. 103, 407; Emilsen 2001, pp. 56–62):

> I tell you that if you will understand, appreciate and act up to the spirit of this passage, you won't even need to know what place Jesus or any other teacher occupies in your heart. If you will do the proper scavenger's work, clean and purify your hearts and get them ready, you will find that all these mighty teachers will take their places without invitation from us. (Gandhi 1958–1994, p. 35:343)

In a letter to Mirabehn, Gandhi shows how both verse 2:59 from the *Bhagavadgītā* and Mt 6:33 direct us toward God as our highest good, who will deliver us from the fears which come along with mortality:

> Objects of senses are eradicated only by seeing God face to face, in other words by faith in God. To have complete faith in God is to see Him. [ . . . ] When we meet Him, we will dance in the joy of His Presence and there will be neither fear of snakes nor of the death of dear ones. For there is no death and no snake-bites in His Presence. (Gandhi 1958–1994, p. 52:257–58)

Gandhi's emphasis on overcoming the fear of death leads us to investigate the deepest roots of mimetically incited violence. According to Girard, it is a fundamental "lack of being" that pushes human beings to imitate the desire of others, and often leads to rivalries (Girard 1977, p. 146). With the help of the cultural anthropologist Ernest Becker, we can complement Girard's mimetic theory by emphasizing death anxiety as the cause of this lack of being. According to Becker, human mortality causes an existential longing for self-esteem of cosmic significance, which people can only obtain from others. This human inadequacy easily ends up in competitive struggles for recognition and other types of mimetic rivalries. Becker explains with it the "ubiquitousness of envy", and mentions "sibling rivalry" to demonstrate this human predicament (Becker 1975, p. 12; Becker 1997, p. 4):

> Sibling rivalry is a critical problem that reflects the basic human condition: it is not that children are vicious, selfish, or domineering. It is that they so openly express man's tragic destiny: he must desperately justify himself as an object of primary value in the universe; he must stand out, be a hero, make the biggest possible contribution to world life, show that he *counts* more than anything or anyone else.

Becker's anthropological insight not only complements Girard's mimetic theory but addresses existential problems with which all world religions must deal.

For a better understanding of Gandhi's focus on overcoming the fear of death, we can turn to Anantanand Rambachan, a professor of religion, who discovered a fascinating "coincidence of terminology between Becker and the Upaniṣads" (Rambachan 2015, p. 30). *Advaita* understands that the usual worldly means to overcome death anxiety are futile:

> The fundamental human predicament, as understood in Advaita, is that of a self-conscious being experiencing a profound sense of inner lack and insignificance and discovering that culturally approved gains such as pleasure, wealth, fame, and power do not resolve this emptiness. (Rambachan 2015, p. 26)

According to *advaita*, what Becker called a longing for "cosmic significance" is "the intrinsic desire for *brahman* (the infinite), where alone there is freedom from suffering [ . . . ]. The infinite is, according to the Advaita tradition, what human beings really want, as opposed to the unending finite ends that we pursue." (Rambachan 2015, p. 30).

With the help of Rambachan's Hindu theology of liberation we can understand the religious background of Gandhi's emphasis on overcoming the fear of death by opening up to God. We can again recognize an analogy to the Christian distinction between temporal and eternal goods. Furthermore, it is not by chance that the song of Zechariah at the beginning of the Gospel of Luke states that humans who live in the "shadow of death" need the light of God's "tender mercy" to "guide our feet into the way of peace" (Lk 1:78–79). Gandhi, too, saw the overcoming of the fear of death as a prerequisite for the satyagrahis, the nonviolent soldiers. They have to be "free from fear, whether as to their possessions, false honour, their relatives, the government, bodily injuries or death" (Gandhi 1958–1994, p. 10:53). All nonviolent action depends on detachment that ultimately only God's grace can give.

## 6. Conclusions

Studying the life and work of Gandhi with the help of Girard's mimetic anthropology shows first that the Indian satyagrahi, too, recognized many instances of mimetic rivalries causing violence. Like Girard, he was also able to deconstruct racist claims of cultural incompatibilities by referring to jealousy and envy as the real causes of conflicts. Girard and Gandhi also share an insight into the contagious nature of violence. Violence tends to escalate, and has led in our modern world to nuclear rivalries threatening the survival of humanity. Recognizing the dangers of violence, both Girard and Gandhi understand the importance of nonviolence for the avoidance of humanity's self-destruction. They share an appreciation of the nonretaliation that is expressed in the Sermon on the Mount. They differ, however, in their interpretation of this seminal text in the New Testament. Girard stands for a more passive renunciation of violence, as his mixing up of the cheek that Jesus mentions illustrates. Gandhi recognized in Jesus an active satyagrahi in close affinity to his militant understanding of nonviolence as a third way between indifferent complacence and the retaliatory mirroring of violence. A final step shows similarities between Girard and Gandhi in their religious understandings of the ways in which to overcome destructive desires. Both recognize the dangers that follow the desire for indivisible goods. Girard's anthropology has a close affinity with the Christian teaching of God as the highest good, and the distinction between temporal and eternal goods. Gandhi's Hindu background shows analogous insights in his readings of the *Bhagavadgītā* and the Upanishads. The deepest roots for mimetic desire can be found in humanity's mortality, which leads human beings to seek their significance competitively. By opening up to God, humans can overcome their

fear of death and create peace by breaking out of mimetic entanglements. Studying Gandhi in the light of Girard's anthropology increases the plausibility of his understanding of nonviolent action. At the same time, mimetic theory is improved by a model of nonviolent practice. The discovery of many parallels in Gandhi's Hindu tradition also broadens the religious scope of Girard's approach.

**Funding:** This research received no external funding.

**Institutional Review Board Statement:** Not applicable.

**Informed Consent Statement:** Not applicable.

**Data Availability Statement:** Not applicable.

**Acknowledgments:** I would like to thank my colleagues Louise du Toit, Ephraim Meir, and Ed Noort who discussed earlier versions of this article during our common research on Gandhi at STIAS in 2021. I am also grateful to the anonymous reviewers for their helpful suggestions and to the doctoral students at the Catholic-Theological Faculty of the University of Innsbruck who read and discussed with me Gandhi's autobiography during winter semester 2019/20. Annette Edenhofer helped me to find passages showing Gandhi's awareness of violent escalation, and Tony Bharath Kenneth Mathew referred me to the *advaita* tradition in India.

**Conflicts of Interest:** The authors declare no conflict of interest.

## Notes

[1] Rivalries and even wars between religions are not uncommon in our world. According to Girard, however, they result from religious actors who substitute the adoration of the holy with worldly pursuits (Palaver 2013, pp. 93–95). Similarly, Gandhi also claims that fights between religions "are not part of religion although they have been practised in its name" (Gandhi 1958–1994, p. 10:24).

[2] Highlighting Gandhi's deconstruction of racism does not mean that he was free of racial prejudices when he came to South Africa (Meer 1995, pp. 1027–41; Kolge 2016). He, however, overcame over the years his own prejudices. In this he was positively influenced by John Dube, the founding president of the African National Congress, who reciprocally appreciated Gandhi's nonviolent struggle against racial discrimination (Reddy 1995, pp. 27–32; Presbey 2016). Nelson Mandela was right when he claimed, in 1995, that the African struggle "is rooted [ . . . ] in the Indian struggle", referring to the influence that Gandhi and Dube had on each other (Mandela 1995, p. 563).

[3] Recent criticisms of Gandhi's relation to the caste system often overlook the complexity of his argument. One can, however, claim that he was most likely rather naïve to believe that the discriminatory caste system could be changed without a strong liberating initiative by the Dalits like Dr. Ambedkar themselves (Dumont 1980, p. 223; Rambachan 2019, p. 155).

[4] More in line with the Sufi tradition of the moth-and-flame metaphor, Gandhi could appreciate in the poem 'Fakirs We'—which he mentions in connection with the poem 'Blow for a Blow'—lines like these: "Fakirs we've made of ourselves/For the motherland's sake;/We've kindled the flame of love/To burn us for India's sake." (Gandhi 1958–1994, p. 9:491). What he appreciates in this poem is the "voluntary poverty" that should characterize every satyagrahi. This attitude differs from the revolutionaries whose sacrifices were more the result of their imitation of colonial violence. Like the revolutionaries, Gandhi too understood that nonviolence requires sacrifices. He distinguished, however, a "pure sacrifice" from the "thoughtless annihilation of the moth in the flame. [ . . . ] Without the requisite purity, sacrifice is no better than a desperate self-annihilation devoid of any merit. Sacrifice must [ . . . ] be willing and it should be made in faith and hope, without a trace of hatred or ill-will in the heart." (Gandhi 1958–1994, p. 85:203). For a more detailed comparison between Gandhi's and Girard's understanding of sacrifice, see Palaver 2019.

[5] Shamal Bhatt's stanza ends with "return with gladness good for evil done" (Gandhi 1958–1994, p. 39:34) and emphasizes, like the Sermon on the Mount or Rom 12:21, that evil should be overcome with good.

[6] It is important to note that Gandhi addresses, here, an exceptional case. In his eyes, good intentions do not justify any means, but have to correspond to ends. If, however, violence cannot be avoided, it is important to refrain from any desire to kill or injure.

[7] We have to distinguish between Gandhi's personal rejection of war and his political insight that as long as a great majority of the people do not follow his example, nonviolence cannot fully be realized. Gandhi hoped for the "adoption of non-violence to the utmost extent possible", at least for a "militarism of a modified character" (Gandhi 1958–1994, p. 76:215–16). This becomes most obvious in his letter to president Roosevelt in July 1942, in which he proposed that the Allies could keep their troops in a free India to prevent Japanese aggression: "My personal position is clear. I hate all war. If, therefore, I could persuade my countrymen, they would make a most effective and decisive contribution in favour of an honourable peace. But I know that all of us have not a living faith in non-violence" (Gandhi 1958–1994, p. 76:264; cf. 186–87, 207–8; Fischer 1953, pp. 425–26; Guha 2018, p. 659; Coovadia 2020, pp. 128–29).

8      I follow Gandhi's translation in his "Discourses on the 'Gita'" (Gandhi 1958–1994, p. 32:94–376).

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
