# Peer review of "Gandhi’s Militant Nonviolence in the Light of Girard’s Mimetic Anthropology"

_religions, doi:10.3390/rel12110988_

Round 1

Reviewer 1 Report

This article demonstrates in a compelling manner that reading Gandhi in light of Girard and Girard in light of Gandhi is a highly productive exercise.  The author is able to show that Girard’s mimetic theory benefits from links to an “active practice of nonviolence” attributable to Gandhi just as Gandhi’s concept of satyagraha receives beneficial support when undergirded by principles of mimetic theory with which it is shown to be highly compatible.  By reinforcing fundamental features of Gandhi’s thought with concepts grounded in Girard’s cultural anthropology that explores human violence broadly across time and in its many cultural expressions the author confirms the cross-cultural salience of Gandhi’s vision and philosophy.  I believe that, in reading Gandhi from a Girardian perspective, this essay either breaks entirely new ground or, at the very least, explores Gandhi’s thought from a perspective uncommon among Gandhi scholars.  Perspectives that illuminate a thinker’s work in in new ways typically invigorate scholarship focused on that thinker; I expect this essay to be received as such by Gandhi scholars.  Likewise, Girard’s rather vague descriptions of what the actual practice of nonviolence looks like are bolstered by Gandhi’s thought and examples from his active practice of nonviolence.  So also does Gandhi attest to religious traditions outside of Christianity that promote and achieve alternatives to cycles of violence in which humans are mired.  Girard’s tendencies to drift into Christian exceptionalism are thereby countered, which is a crucial contribution to new currents in Girardian scholarship that broaden its traditional focus on Christianity (and Judaism by extension) to other world religious traditions in the interest of seeing how nonviolent interventions in human violence are found across the spectrum of the world religions.  Thus, I anticipate that scholars whose perspectives are attuned to Gandhi and those whose work is grounded in mimetic theory will be equally appreciative of the achievements of this significant piece of scholarship. 

This is an extremely timely article that engages current issues of violence, racism, colonialism while enhancing readers’ appreciation for the ongoing viability of Gandhi and Girard as important voices in discussions of these issues.  The argument is complex but clearly written.  It is grounded in a careful reading of texts and a current bibliography.  Its comparative approach is extremely effective in illuminating the work of both figures.  My specific comments are suggestions to the author on minor revisions that will enhance communication of the argument to the readers.  The essay should be published with dispatch as soon as the author can address reader responses.

Author Response

Many thanks for your very careful reading of my text and your many helpful suggestions. I really appreciate your effort and I tried – as far as possible – to take up all your specific suggestions in the revised version.

Reviewer 2 Report

Lines 84-86: What is about rivalries between religions?

Line 321: "months in London" instead of "months In London"

Lines 543-547: Isn't there a contradiction or at least a tension to the principle that means have to correspond to ends? Does good intention justify each kind of means? 

Author Response

Many thanks for reading my article and for addressing three problems. I corrected the typo in line 321 and inserted two footnotes to address your other two concerns.

Reviewer 3 Report

This is a very high quality article. The author demonstrates a thorough mastery of the literature from and adjacent to both Girard and Gandhi, as well as wider literature on active nonviolence and the religious traditions that have contributed to it historically. The organization of the article is clear and the prose is extremely well polished. As a reviewer, I don't recall ever having so little to say by way of suggestions. Indeed, I have only one tiny editing suggestion: The one place I stumbled in reading the piece was at lines 714-715. Is there a typo or something missing here? Even if not, perhaps the author can smooth out the prose a bit. 

Author Response

Many thanks for reading my article and for your appreciative review. I rephrased lines 714-715 that you mentioned as a stumbling block.

Reviewer 4 Report

I think this piece would make a significant contribution to the field, with its careful correlation of Gandhian nonviolence with Girard's work, illuminating the nuances of each scholar's work in the process.

If published, I would use it in my course on nonviolence as ascetic discipline.

Author Response

Many thanks for reading the text and for your appreciation of this article. I am glad to hear that you may use it in your course on nonviolence.